# Aberrant Expression of *COX-2* and *FOXG1* in Infrapatellar Fat Pad-Derived ASCs from Pre-Diabetic Donors

**DOI:** 10.3390/cells11152367

**Published:** 2022-08-01

**Authors:** Benjamen T. O’Donnell, Tia A. Monjure, Sara Al-Ghadban, Clara J. Ives, Michael P. L’Ecuyer, Claire Rhee, Monica Romero-Lopez, Zhong Li, Stuart B. Goodman, Hang Lin, Rocky S. Tuan, Bruce A. Bunnell

**Affiliations:** 1Center for Stem Cell Research and Regenerative Medicine, Tulane University School of Medicine, New Orleans, LA 70112, USA; bodonne1@tulane.edu (B.T.O.); tmonjure@tulane.edu (T.A.M.); sara.al-ghadban@unthsc.edu (S.A.-G.); cives1@tulane.edu (C.J.I.); mlecuyer@tulane.edu (M.P.L.); 2Departments of Orthopaedic Surgery and Bioengineering, Stanford University School of Medicine, Stanford, CA 94305, USA; cr21@stanford.edu (C.R.); monik.rom@gmail.com (M.R.-L.); goodbone@stanford.edu (S.B.G.); 3Center for Cellular and Molecular Engineering, Department of Orthopaedic Surgery, University of Pittsburgh School of Medicine, Pittsburgh, PA 15213, USA; alanzhongli@pitt.edu (Z.L.); hal46@pitt.edu (H.L.); tuanr@cuhk.edu.hk (R.S.T.)

**Keywords:** osteoarthritis, COX-2, FOXG1, adipose stem cell, infrapatellar fat pad, diabetes

## Abstract

Osteoarthritis (OA) is a degenerative joint disease resulting in limited mobility and severe disability. Type II diabetes mellitus (T2D) is a weight-independent risk factor for OA, but a link between the two diseases has not been elucidated. Adipose stem cells (ASCs) isolated from the infrapatellar fat pad (IPFP) may be a viable regenerative cell for OA treatment. This study analyzed the expression profiles of inflammatory and adipokine-related genes in IPFP-ASCs of non-diabetic (Non-T2D), pre-diabetic (Pre-T2D), and T2D donors. Pre-T2D ASCs exhibited a substantial decrease in levels of mesenchymal markers CD90 and CD105 with no change in adipogenic differentiation compared to Non-T2D and T2D IPFP-ASCs. In addition, Cyclooxygenase-2 (*COX-2*), Forkhead box G1 (*FOXG1*) expression and prostaglandin E2 (PGE_2_) secretion were significantly increased in Pre-T2D IPFP-ASCs upon stimulation by interleukin-1 beta (IL-1β). Interestingly, M1 macrophages exhibited a significant reduction in expression of pro-inflammatory markers *TNFα* and *IL-6* when co-cultured with Pre-T2D IPFP-ASCs. These data suggest that the heightened systemic inflammation associated with untreated T2D may prime the IPFP-ASCs to exhibit enhanced anti-inflammatory characteristics via suppressing the IL-6/COX-2 signaling pathway. In addition, the elevated production of PGE_2_ by the Pre-T2D IPFP-ASCs may also suggest the contribution of pre-diabetic conditions to the onset and progression of OA.

## 1. Introduction

Symptomatic osteoarthritis (OA) in the knee is one of the leading global causes of physical disability. It affects approximately 14 million people in the USA and millions more across the globe [1,2]. The knee joint comprises multiple tissues, including cartilage, subchondral bone, synovial membrane, and the infrapatellar fat pad (IPFP). OA results from the disruption of cartilage homeostasis and osteophyte formation within the synovial capsule [3]. Disease pathology extends beyond the articular cartilage and subchondral bone to include chronic inflammation. Thus, OA is described as a whole joint disorder [4,5,6]. Furthermore, age-associated overuse or traumatic injury are often the initiators of OA [7,8]. Still, long-term inflammation is often highlighted as the driving force behind ECM degradation [9].

A primary hallmark of OA is the presence of chronic inflammation within the synovial joint and surrounding tissues [4,6,10]. While the exact tissue source(s) and the composition of the proinflammatory factors in knee OA are still under investigation, focusing primarily on the synovium, other joint tissues, such as the IPFP, are also hypothesized to play a critical role [10]. Increased levels of proinflammatory cytokines, in particular interleukin (IL)-1β, IL-6, and tumor necrosis factor α (TNFα), are found in the synovial fluid of OA patients [11,12]. Moreover, IL-1β and TNFα stimulate their production and induce the expression of IL-6, IL-8, and prostaglandin production by chondrocytes [13,14]. The increased expression of mediators of inflammation induces the expression of matrix metalloproteases (MMPs) in chondrocytes. The MMPs drive the destruction of the ECM, as indicated by elevated collagen and proteoglycan breakdown [13,15].

Additionally, inflammation inhibits the chondrocytes’ production of collagen type II and proteoglycans. Most patients with knee OA also exhibit synovitis, suggesting synovial inflammation is central in OA progression [16]. The synovium is a source of circulating immune cells that direct cartilage breakdown in OA. While inflammation of the cartilage and synovium is well characterized in OA, it is not evident that the high levels of proinflammatory mediators found in the synovial fluid of OA patients result from production by the synovium and chondrocytes exclusively. Many OA patients have detectable inflammation in other joint tissues. The IPFP may be one potential source of proinflammatory mediators in knee OA [17].

Studies have demonstrated that adipose tissue reservoirs throughout the body are active endocrine production sites that contribute to systemic inflammation [18]. Patients diagnosed with type II diabetes mellitus (T2D) have elevated IL-6 and TNFα in their serum [19]. T2D is classified as a weight-independent risk factor for OA, suggesting a link between T2D-mediated adipose dysfunction and OA [20,21]. OA patients also exhibit elevated synovial concentrations of adipokines, such as adiponectin (ADIPOQ) and leptin (LEP) [22,23]. When exposed to LEP, cartilage explants and isolated chondrocytes demonstrated heightened expression of IL-1β, MMP-9, and MMP-13 [23,24]. Cartilage does not produce ADIPOQ or LEP, making it unclear what tissue is responsible for their increased concentration in synovial fluid. Due to its adjacency to the synovial fluid, the IPFP could be the source of OA modifying cytokines; however, to our knowledge, this direct link has not been established, nor has the IPFP from T2D patients been investigated.

As an extension of this concept, stem/progenitor cells isolated from adipose depots have also shown variation in differentiation potential and inflammation-related properties [25]. Due to their inherent anti-inflammatory properties and heightened chondrogenic potential compared to subcutaneous derived ASCs, ASCs derived from IPFP (IPFP-ASCs) are often highlighted as a potential therapeutic cell source for OA [26,27]. However, treatment of OA with the injection of ASCs has resulted in mixed outcomes, often demonstrating only transient therapeutic benefits, such as pain alleviation and improved mobility [28]. Therefore, it is plausible to think that the characteristics of the ASC donor may influence their therapeutic benefits. In general, stem cell differentiation potential and immunomodulation properties are susceptible to donor physiologic factors, such as tissue site, age, obesity, disease (e.g., T2D) and gender [25,29,30,31,32]. While IPFP-ASCs from OA patients maintain their chondrogenic differentiation potential, the impact of T2D on their differentiation potential or their immunomodulatory properties is unknown [33].

Chronic inflammation of the IPFP, often associated with OA, is characterized by fibrosis, hyperplasia and neural ingrowth [17,34,35]. However, little is known about the inflammatory microenvironment of OA IPFP and the effect of intercellular interaction and communication (direct or paracrine) on ASC differentiation potential. Studies have shown that IPFP-ASCs have increased chondrogenic potential compared to subcutaneous ASCs and no OA impact on the chondrogenic potential of IPFP-ASCs has been noted [36,37]. However, another study by Stocco et al. demonstrated changes in the immunotypic profile of IPFP-ASCs isolated from OA patients, suggestive of an impaired ability by IPFP-ASCs to modulate OA-related inflammation. To our knowledge, no study has investigated the combined impact of OA and T2D on IPFP-ASCs. Thus, investigating the impact of T2D on the biologic properties of IPFP and IPFP-derived ASCs is essential for understanding their therapeutic potential in the setting of OA.

The goals of the research studies presented in this paper were to (a) characterize the stemness properties of IPFP-ASCs isolated from non-diabetic (Non-T2D), pre-diabetic (Pre-T2D), and diabetic (T2D) patients and (b) determine the levels of expression of inflammatory mediators and adipokines in IL-1B stimulated IPFP-ASCs and adipocytes and in a co-culture system with monocyte-derived macrophages.

## 2. Materials and Methods

### 2.1. IPFP-ASC Isolation and Expansion

A total of nine IPFP explants were collected from patients undergoing total knee arthroplasty at Stanford Hospital. The samples were collected with Institutional Review Board approval and the patient’s knowledge and consent. The patient characteristics and diabetic status are listed in Table 1. Their primary care physician determined the diabetic status of each donor. IPFP-ASCs were isolated as previously described [38]. Briefly, IPFP samples were minced in 30 mL sterile 1X phosphate-buffered saline (PBS, Hyclone, Logan, UT, USA). An equal volume of digestion solution, 0.1% type I collagenase (Worthington Biochemical, Lakewood, NJ, USA) and 1% bovine serum albumin (Sigma-Aldrich, St. Louis, MO, USA) in 1X PBS, was added and samples were incubated at 37 °C. After one hour, growth medium consisting of Dulbecco’s Modified Eagle’s Medium Nutrient Mixture F-12 Ham (1:1 mixture) (DMEM/F-12, Gibco, Gaithersburg, MD, USA), supplemented with 10% fetal bovine serum (FBS, Hyclone, Logan, UT, USA) and 1% antibiotic/antimycotic (Anti/Anti, Gibco) was added to the samples. After centrifugation at 300× *g* for 5 min, the resulting cell pellets were suspended in growth medium and plated on 150 cm^2^ Nunc plates (ThermoFisher Scientific, Pittsburgh, PA, USA). After 48 h, the plates were washed with 1X PBS, and the adherent cells were expanded in growth medium to 70% confluency before being passaged at 400 cells/cm^2^. Sub-confluent ASCs (<~70%) from passages 3–4 were used for all experiments.

### 2.2. Flow Cytometry

The expression of ASC cell surface markers was assessed by flow cytometry. IPFP-ASCs were washed in 1X PBS before being blocked in Fc block (ThermoFisher Scientific). Cells were immunostained for 30 min. with conjugated antibodies against CD90 (Ext/Em: 488/530) (Cat #: 11-0909-42) (EBioscience, San Diego, CA, USA), CD73 (Ext/Em: 488/560) (Cat #: 550257) (BD Pharmingen), CD105 (Ext/Em: 633/670) (Cat #: 17-1057-41) (EBioscience), CD3 (Ext/Em: 488/630) (Cat #: 61-4724-82) (EBioscience), CD14 (Ext/Em: 488/670) (Cat #: IM2640U) (Beckman Coulter), CD31 (Ext/Em: 488/770) (Cat #: 25-0349-41) (EBioscience), CD45 (Ext/Em: 630/710) (Cat #: A71117) (Beckman Coulter). All antibodies were diluted 1:1000 from commercial concentrations. Samples were analyzed with a Gallios Flow Cytometer (Beckman Coulter, Brea, CA, USA) with Kaluza software (Beckman Coulter). A minimum of 10,000 events were captured and analyzed for each sample.

### 2.3. Colony-Forming Unit Assay (CFU)

The ASCs were analyzed for self-renewal activity by a colony-forming unit fibroblast (CFU-F) assay. The cells were seeded onto a 6-well NUNC cell culture plate at 100 cells/well. The cells were cultured in growth medium for 14 days, with the medium being refreshed once after seven days. After 14 days, the wells were washed with 1X PBS and stained with 3% (*w*/*v*) crystal violet (Sigma-Aldrich). To monitor the number of cells capable of self-renewal, the number of colonies greater than 3 mm^2^ in diameter was counted and divided by the total number of cells seeded.

### 2.4. Osteogenic and Adipogenic Differentiation

Each IPFP line was differentiated along adipogenic and osteogenic lineages to assess the differentiation potential of each line. Briefly, 20,000 cells/cm^2^ were seeded onto a 6-well NUNC cell culture plate. The cells were allowed to reach 100% confluency in the growth medium before being replaced with an osteogenic or adipogenic differentiation medium (AdipoQual (AQ); Obatala Sciences, New Orleans, LA, USA). Osteogenic Media (OM) consisted of 89% DMEM-F12, 10% FBS, and 1% Anti/Anti supplemented with 10 nM Dexamethasone, 20 mM β-glycerophosphate, 50 µM L-Ascorbic Acid 2-phosphate (Sigma-Aldrich). The differentiation medium was replaced every 3–4 days for 28 days. 

After differentiation in adipogenic medium, each well was washed with 1X PBS, fixed for 1 h in 4% Paraformaldehyde (PFA) and then stained in a 0.5% filtered solution of Oil Red O (Sigma-Aldrich) for 10 min. Oil Red O was extracted from the cultures using 100% isopropanol to quantify staining. The absorbance at 584 nm was measured using a Synergy^TM^ HTX Multi-Mode Microplate Reader (BioTek, Winooski, VT, USA). Absorbance values were normalized to total protein content and determined using a Pierce BCA kit (ThermoFisher Scientific). RNA and conditioned media were collected and stored at −80 °C·for future analysis.

After differentiation in osteogenic media, each well was washed with deionized water and fixed for 1 h in 4% PFA before being stained with 5% Alizarin Red for 20 min (Sigma-Aldrich). Images were then acquired at 20× magnification on a Nikon Eclipse TE200 microscope equipped with Nikon Digital Camera DXM1200F and Nikon ACT-1 software version 2.7 (Nikon, Melville, NY, USA). 

### 2.5. Co-Culture of ASCs with Monocyte-Derived Macrophages

Human whole blood was purchased from the Stanford Blood Center. Monocytes were isolated from these samples using EasySep Human Monocyte Isolation (STEMCELL Technologies, Vancouver, BC, Canada), according to the manufacturer’s protocol. Macrophage differentiation of monocytes was carried out by plating the cells at 1 × 10^5^ cells/cm^2^ in the bottom well of a Corning Transwell and culturing them in growth medium supplemented with 100 ng/mL macrophage colony-stimulating factor (MCSF, R&D Systems, Minneapolis, MN, USA) for six days. A pro-inflammatory (M1) phenotype was induced by treating the macrophages with both 20 ng/mL interferon-gamma (INF-γ, R&D systems and 10 ng/mL lipopolysaccharides (LPS, R&D systems) for 48 h. An anti-inflammatory (M2) phenotype was induced by treating the macrophages with 20 ng/mL IL-4 (R&D Systems). All experiments were conducted with pooled IPFP ASC lines from three patients. A total of 0.25 million IPFP-ASCs were seeded on the 0.4 μm pore insert of Transwell culture plates and allowed to acclimate for 24 h in growth medium before the inserts were placed in co-culture with the M0, M1, or M2 macrophages. Then, the co-cultures were cultured for 72 h before the media and RNA were collected from the IPFP-ASCs and the macrophages for analysis by RT-PCR.

### 2.6. IL-1β Induction 

IPFP-ASCs were cultured to 70% confluency in growth medium for six days, with the medium refreshed after three days. The medium was then replaced with growth medium supplemented with 10 ng/mL IL-1β and the cells were cultured for an additional 24 h (PeproTech, Rocky Hill, NJ, USA). RNA and conditioned media were collected for future analysis.

### 2.7. Quantitative Polymerase Chain Reaction (qPCR)

RNA extraction of all samples was done using RNeasy Microkit (Qiagen, Germantown, MD, USA) according to the manufacturer’s protocol. Then, reverse transcription was done with 1 μg of RNA using a High-Capacity cDNA Reverse Transcription Kit containing RNase Inhibitor (ThermoFisher Scientific).

qPCR was performed with SsoAdvanced Universal SYBR Green Supermix (Bio-Rad, Hercules, CA, USA) with custom primers purchased from Integrated DNA Technologies (IDT, Newark, NJ, USA). Gene-specific primer sequences are listed in Table 2. The fold change in gene expression was calculated using the -∆∆CT method and the expression of the glyceraldehyde 3-phosphate dehydrogenase (*GAPDH*) was used as a housekeeping gene. The reference group for all experiments was Non-T2D IPFP-ASCs.

### 2.8. Enzyme-Linked Immunosorbent Assay (ELISA)

The conditioned cell culture media samples were cleared by centrifugation at 1000× *g* for 1 min and stored at −80 °C until use. Then, the media samples were brought to room temperature before analysis using a PGE_2_ ELISA kit according to the manufacturer’s protocols (Pierce BCA kit, ThermoFisher Scientific). Absorbance at 450 nm was measured using a Synergy^TM^ HTX Multi-Mode Microplate Reader. The ELISA data were expressed as a function of total protein content.

### 2.9. Statistical Analysis

Statistical analyses were performed using GraphPad 8.3 software (GraphPad). Statistical significance between 7-day confluent control and experimental groups was determined using two-tailed Student’s *t*-tests. Statistical significance between non-T2D, pre-T2D, and T2D experimental groups was determined using ordinary one-way ANOVA with Tukey posthoc analysis. All data represent the mean ± standard error, and all experiments were performed in triplicates. Significance was assessed as *p* < 0.05 and is denoted by *, with ** for *p* < 0.01, *** for *p* < 0.001, and **** for *p* < 0.0001.

## 3. Results

### 3.1. Altered Cell Surface Antigen Expression in Pre-T2D IPFP ASCs

Standard ASC characterization methods were used to assess the stemness of IPFP-ASCs. All groups of IPFP-ASCs exhibited fibroblastic cell morphology similar to subcutaneous ASCs (data not shown) after the first passage. The profiling of cell surface markers by flow cytometry demonstrated robust expression of CD90 and CD105 (~95%) and reduced CD73 (~75%) expression in Non-T2D and T2D IPFP-ASCs (Figure 1A) along with undetectable expression of CD3, CD14, CD31, and CD45 cell surface markers (Figure 1A, Appendix A). Pre-T2D IPFP-ASCs demonstrated a significant reduction in CD90 and CD105 compared to Non-T2D and T2D IPFP-ASCs without a corresponding increase in CD3, CD14, CD31, or CD45 expression (Figure 1A, Appendix A). 

Furthermore, Pre-T2D IPFP-ASCs tended to exhibit increased CFU activity compared to Non-T2D IPFP-ASCs, while T2D IPFP-ASCs showed significantly elevated CFU activity compared to Non-T2D IPFP-ASCs (Figure 1B). Adipogenic differentiation of all samples for 28 days resulted in the expected increase in expression of peroxisome proliferator-activated receptor (*PPARγ*), glucose transporter type 4 (*GLUT4*), adiponectin (*ADIPOQ*), and perilipin *(PL1N)* (Figure 1C–F). No differences in the expression levels of *PPARγ*, *GLUT4*, *ADIPOQ*, *or PL1N* were noted among the groups. Oil Red O staining of adipogenic differentiated IPFP-ASCs illustrated accumulation of neutral lipid droplets after culture for 28 days (Figure 2A–C). Quantification of the Oil Red O staining revealed no significant differences in the levels of lipid accumulation among adipocytes differentiated from Non-T2D, Pre-T2D, and T2D cells (Figure 2D). In addition, there were no notable differences in osteogenic differentiation between Non-T2D, Pre-T2D, and T2D sample groups (Appendix A).

### 3.2. Decreased COX-2 and IL-6 Gene Expression in Adipogenically Differentiated IPFP-ASCs 

Analysis of the levels of gene expression for inflammatory mediators and diabetes-associated signaling factors, including *COX-2*, chemokine ligand 5 (*CCL-5*), *IL-6*, *LEP*, intercellular adhesion molecule 1 (*ICAM-1*), insulin-like growth factor-binding protein 5 (*IGFBP5*), insulin receptor substrate 2 *(IRS-2),* forkhead box G1 *(FOXG1)*, and *IL-33* in Non-T2D, Pre-T2D, and T2D IPFP-ASCs demonstrated significant increases in the mRNA levels for both *COX-2* and *CCL-5* in undifferentiated Pre-T2D samples compared to non-T2D and T2D samples (Figure 3A,B). After adipogenic differentiation, a significant decrease in *IL-6* and *COX-2* gene expression was noted in all groups compared to their undifferentiated state (Figure 3A,C). Substantial reductions in mRNA expression levels were detected for *LEP*, *CCL-5*, *ICAM-1*, and *IGFBP5*, and an increase in *IRS-2* in differentiated Non-T2D IPFP-ASCs compared to Non-T2D IPFP-ASCs (Figure 3B,D–G) was detected. Interestingly, *FOXG1* gene expression increased in undifferentiated and differentiated Pre-T2D IPFP-ASCs compared to Non-T2D and T2D ASCs (Figure 3H). Finally, significantly suppressed expression of both *IL-33* and *IGFBP5* was noted in differentiated T2D IPFP-ASCs compared to undifferentiated T2D-IPFP ASCs (Figure 3F,I). 

### 3.3. Increased COX-2, IL-6 and FOXG1 Gene Expression in IL-1β Stimulated Pre-T2D IPFP-ASCs

Exposure of ASCs to IL-1β is a standard technique to induce proinflammatory degenerative features in vitro models. It was used in these studies to simulate an OA-like inflammatory environment [39]. After IL-1β stimulation, IPFP-ASCs exhibited significant increases in the expression of both *IL-6* and *COX-2* in all three groups compared to their respective unstimulated counterparts (Figure 4A,C). Additionally, there was a significant increase in *IL-1β* expression and a decrease in *TNFα* expression in IL-1β stimulated Non-T2D and T2D IPFP-ASC sample groups (Figure 4D,E). T2D IPFP-ASCs exhibited a significant increase in the expression of *IL-33* compared to their undifferentiated controls (Figure 4I). Interestingly, the expression of *IL-10*, considered a potent anti-inflammatory cytokine, was significantly reduced in IL-1β stimulated Non-T2D IPFP-ASCs compared to IPFP-ASC control. However, IL-1β significantly stimulated *IL-10* expression in T2D IPFP-ASCs compared to their non-IL-1β treated controls (Figure 4F). Furthermore, after IL-1β stimulation, there was a significant increase in *IL-10* expression in T2D IPFP-ASCs compared to Non-T2D and Pre-T2D IPFP-ASCs (Figure 4F). Expression levels of *COX-2*, *IRS-2*, and *FOXG1* were also significantly elevated in IL-1β stimulated Pre-T2D IPFP-ASCs compared to IL-1β stimulated Non-T2D and T2D IPFP-ASCS (Figure 4A,G,H). In addition, *TNFα* expression was significantly increased in IL-1β stimulated Pre-T2D IPFP-ASCs compared to Non-T2D IPFP-ASCs (Figure 4D). 

### 3.4. Reduced Expression of TNFα and IL-6 in M1 Macrophages Co-Cultured with Pre-T2D IPFP-ASCs

The activity of synovial macrophages is a crucial component in OA progression, and ASCs are known to modulate their activity [4]. Decreased levels of expression of both *TNFα* and *IL-6* mRNAs were detected in M1 macrophages co-cultured with Pre-T2D and T2D IPFP-ASCs (Figure 5A,C). However, when co-cultured with Pre-T2D IPFP-ASCs, *TNFα* and *IL-6* mRNA expression levels in M1 cells were equivalent to the levels observed in M0 and M2 macrophages (Figure 5A,C). Conversely, there was a significant decrease in *IL-10* gene expression by M1 macrophages co-cultured with Pre-T2D IPFP-ASCs, compared to co-cultured M0 or M2 macrophages (Figure 5D). 

### 3.5. Decreased Expression of Leptin and IL-10 and Increased Expression of COX-2 and FOXG1 in Pre-T2D IPFP-ASCs Co-Cultured with M1 Macrophages

When co-cultured with M1 macrophages, Pre-T2D and T2D IPFP-ASCs exhibited a significant decrease in *LEP* expression and a significant increase in *IL-6* expression compared to naïve IPFP-ASCs (Figure 6C,E); Pre-T2D IPFP-ASCs also demonstrated a significant decrease in *IL-10* and *IRS-2* expression compared to naïve IPFP-ASCs (Figure 6B,G). Additionally, *IL-10* expression levels in Pre-T2D IPFP-ASCs decreased significantly compared to Non-T2D and T2D IPFP-ASCs when co-cultured with M1 macrophages (Figure 6B). The Non-T2D IPFP-ASCs and T2D IPFP-ASCs showed a significant decrease in *TNFα* expression compared to their naïve control when co-cultured with M1 macrophages (Figure 6D). *COX-2* expression by Non-T2D, Pre-T2D, and T2D IPFP-ASCs was significantly increased after co-culture with M1-induced macrophages (Figure 6A). Moreover, *ICAM1* expression was significantly elevated in Non-T2D IPFP-ASCs co-culture with M1 macrophages (Figure 6F). *FOXG1* expression was increased in Pre-T2D IPFP-ASCs compared to Non-T2D and T2D IPFP ASCs (Figure 6H). After co-culture with M2-induced macrophages, IPFP-ASCs exhibited a significant decrease in *TNFα*, *IL-1β*, and *IL-6* (Appendix A).

### 3.6. Increased PGE_2_ Secretion by Pre-T2D IPFP-ASCs

*COX-2*, a synthase for PGE_2_, is a common target for nonsteroidal anti-inflammatory drugs (NSAIDs) prescribed to OA patients [39,40]. A significant decrease in PGE_2_ levels was detected in the conditioned medium of T2D IPFP-ASCs, whereas Non-T2D IPFP-ASCs and Pre-T2D IPFP-ASCs showed increased PGE_2_ production (Figure 7A). Adipogenic differentiation of the three cell types revealed no significant difference in the production of PGE_2_ (Figure 7B). However, IL-1β stimulation of Pre-T2D IPFP-ASCs showed significantly more PGE_2_ production than stimulated Non-T2D and T2D IPFP-ASCs (Figure 7C). No significant differences were detected when co-cultured with M1 macrophages; however, Pre-T2D IPFP-ASCs showed higher concentrations (Figure 7D). All samples co-cultured with M2-induced macrophages had undetectable levels of PGE_2_ (data not shown).

## 4. Discussion

The data presented in this study demonstrate that IPFP-ASCs derived from Pre-T2D patients have a distinct expression of the pattern of *COX-2* when compared to IPFP-ASCs isolated from Non-T2D and T2D patients. When co-cultured with M1 macrophages, Pre-T2D IPFP-ASCs demonstrated an increase in *IL-6* gene expression compared to Non-T2D and T2D IPFP-ASCs. In addition, M1 macrophages showed reduced gene expression of *TNFα* and *IL-6* after exposure to Pre-T2D IPFP-ASCs. Traditionally, samples collected from T2D patients represent the impact of type II diabetes mellitus on IPFP-ASCs [41]; however, a requirement for knee replacement surgery is that each patient must have their diabetes well controlled before surgery. Our data suggest that drugs administered to treat T2D could potentially impact experimental outcomes. Goldman et al. found in a long-term patient study that lifestyle changes and Metformin, a commonly prescribed treatment for T2D patients, are associated with decreases in adiponectin, leptin, and IL-6 serum levels [42]. 

Furthermore, Jenkins et al. found that serum IL-10 and LEP levels decreased in a T2D rat model after exercise treatment and delivery of Metformin [43]. Commonly, preparation for total knee arthroplasty for T2D patients entails an aggressive treatment program to bring their T2D symptoms under control, which raises concern over the effect of treatment on the inflammatory properties of ASCs. Therefore, we included a third sample group of patients diagnosed as pre-diabetic by their primary care physicians. This group was included because they have been diagnosed with heightened serum glucose levels, a potential proinflammatory condition, but have not yet received extensive treatments for T2D.

Most of the disparate properties of IPFP-ASC were described when comparing the Pre-T2D samples and the other two groups. While the cell surface markers CD73, CD90, and CD105, are expressed by ASCs, the non-T2D and T2D cell populations exhibited robust CD73, CD90 and CD105. However, a significant reduction in CD90 and CD105 was detected in the pre-T2D cells without a corresponding increase in other markers. Nawrocka et al. demonstrated that T2D ASCs isolated from subcutaneous adipose tissue (ScAT-ASCs) have similar expression of CD90, CD73, and CD105 to Non-T2D ScAT-ASCs [44]. However, Ferrer et al. found a decrease in CD90, CD73, and CD105 of ScAT-ASCs isolated from a murine T2D model [45]. This is the first investigation into CD90, CD73, and CD105 expression of human ASCs derived from adipose collected from Pre-T2D patients. We hypothesize that changes in cell surface markers by Pre-T2D IPFP-ASCs could result from microenvironmental changes in the adipose tissue depot due to the onset of T2D. The decrease in CD105 expression has been associated with the differentiation of ASCs [45]. Thus, changes in the cellular composition of the IPFP after T2D initiation could partially be accounted for by IPFP-ASC differentiation. The sub-population of Pre-T2D IPFP-ASCs negative for CD90, CD73, and CD105 could be in the beginning stages of differentiation. Bi-lineage differentiation assays revealed no notable differences in adipogenic or osteogenic differentiation potential of Pre-T2D IPFP-ASCs compared to Non-T2D IPFP-ASCs. However, a decrease in differentiation potential may have been detected if the changes in cell surface markers indicated the beginning stages of differentiation.

Notably, *COX-2* expression was increased in Pre-T2D IPFP-ASCs compared to Non-T2D and T2D IPFP-ASCs in an unstimulated state and again after IL-1β stimulation. The COX-2 enzyme is responsible for PGE_2_ synthesis [46,47]. When stimulated with IL-1β, PGE_2_ concentrations in Pre-T2D conditioned media were significantly higher as compared to media conditioned by Non-T2D and T2D IPFP-ASCs. Human cartilage explants exhibited a PGE_2_ dose-dependent response towards the elevated expression of MMP-13 and ADAMTS-5, typically found at elevated levels in OA synovial fluids [48,49]. For this reason, COX-2 inhibitors such as *celecoxib* are common medications prescribed to OA patients to reduce local inflammation, and PGE_2_ function is largely considered catabolic in cartilage tissue [47,48]. The increased expression of *COX-2* and concentration of PGE_2_ by Pre-T2D IPFP-ASCs suggests heightened concentrations of PGE_2_ in the IPFP that would diffuse into the joint space. Long-term disruptive synovial PGE_2_ concentration in Pre-T2D patients could induce a more catabolic state within cartilage and increase the risk of OA after joint injury.

Synovial macrophages play an essential role in cartilage catabolism and the persistence of the inflammatory state within cartilage in OA [4,50]. Macrophages are broadly classified into three functional categories, a resting (M0) phenotype, a pro-inflammatory (M1) phenotype and an anti-inflammatory (M2) phenotype [51]. It is commonly believed that OA patients have increased M1 activity by synovial macrophages, which may be linked to aberrant immunomodulation by IPFP-ASCs in T2D patients. When co-cultured with M1 macrophages, Pre-T2D IPFP-ASCs exhibited increased expression of *IL-6* and a decrease in *IL-10*, suggesting they promote increased M1 induction of macrophages in an inflammatory environment. In fact, after co-culture with Pre-T2D IPFP-ASCs, M2 macrophages exhibited an altered cytokine expression pattern towards an M1 phenotype. In addition, PGE_2_ expression has been found to increase the anti-inflammatory properties of ASCs through an *IL-6-dependent* pathway [52,53]. Although *COX-2* expression levels were not found to be different between Non-T2D, Pre-T2D, and T2D IPFP ASCs co-cultured with M1 macrophages, there was a significant increase in *IL-6* expression. The adjacency of the IPFP to the synovium has led to the belief that the IPFP and the synovium may act as a morpho-functional unit, as reviewed by Macchi [54].

Furthermore, Stocco et al. demonstrated increased expression of HLA-DR, Fas, and FasL by IPFP-ASCs isolated from OA patients, indicating a response of the IPFP to synovial inflammation [55]. We hypothesize that IPFP-ASCs are primed toward increasing *COX-2* expression in Pre-T2D patients by exposure to chronic low-grade inflammation associated with diabetes. While higher *COX-2* expression increases the anti-inflammatory potential of IPFP-ASCs, the elevated PGE_2_ concentrations in the IPFP may induce a more catabolic state in cartilage. Future work will focus on balancing *COX-2* expression in IPFP-ASC and synovial PGE_2_ activity. 

Pre-T2D IPFP ASCs also demonstrated a robust increase (fold change >1000) in *FOXG1* expression compared to Non-T2D IPFP ASCs in all experimental conditions, reaching significance after *IL-1β* stimulation. *FOXG1* is a known inhibitor of *FOXO* and inhibits the PI3K pathway, a major downstream pathway of insulin receptors [56,57]. Loss of *FOXO* expression has been implicated in the maintenance of insulin resistance in T2D patients, as reviewed by Maiese et al., but little is known about the mechanism of action [58]. Because the *FOXG1/FOXO* mode of action in diabetes has yet to be elucidated, it is difficult to say what functional role *FOXG1* plays in IPFP-ASCs. However, increased *FOXG1* expression does imply that Pre-T2D IPFP-ASCs are exhibiting a diabetes-related genotype. In addition, there is evidence of nerve ingrowth into the cartilage and osteochondral junction in mid to late-stage OA, possibly in an attempt to regulate cartilage degradation [59]. The IPFP also contains nociceptive receptors responsive to *IL-1β*, *IL-6*, and *TNFα* [34,60]. *FOXG1* expression in ASCs is commonly studied as a transcription factor for neuronal differentiation [61]. Thus, the upregulation of FOXG1 may suggest an increase in neuronal Pre-T2D IPFP-ASC differentiation potential that could account for increased pain and sensitivity experienced by OA patients.

These studies have established critical transcriptional differences between Non-T2D IPFP-ASCs, Pre-T2D IPFP-ASCs, and T2D IPFP-ASCs that suggest the Pre-T2D condition increases the sensitivity of IPFP-ASCs to a proinflammatory environment. However, many differences between Non-T2D IPFP-ASCs and Pre-T2D IPFP-ASCs were not found in T2D IPFP-ASCs. As noted above, typical T2D treatments have successfully reduced serum levels of LEP and IL-6. Similarly, T2D therapies appear to be effective in returning the expression of LEP and IL-6 by IPFP-ASCs to levels found in Non-T2D, suggesting that earlier T2D intervention may be required, especially in patients who also exhibit OA symptoms.

While all cell types exhibited clonogenic ability, the T2D IPFP-ASCs had significantly increased clonogenic efficiency than the Non-T2D IPFP-ASCs. Nawrocka et al. demonstrated that T2D ScAT-ASCs had suppressed clonogenic expression more than Non-T2D Sc-AT-ASCs. In contrast, Cramer et al. found that long-term high glucose exposure does not significantly impact the clonogenic ability of ASCs [29]. Previously, the same group demonstrated that exposure to basic fibroblast growth factor (bFGF) ameliorates T2D-associated changes in clonogenic ability by ScAT-ASCs; however, bFGF was not investigated in this study [44] Other limitations include (1) lack of information from the clinician about the patient’s treatment which might influence the stemness of IPFP-ASCs and (2) all cells were isolated from patients undergoing knee arthroplasty in response to severe OA symptoms. (3) OA-associated synovial inflammation may have obscured potential differences between Non-T2D, Pre-T2D, and T2D patients due to the overlap of inflammatory pathways between T2D and OA.

The data presented here demonstrate both catabolic and anabolic alterations in Pre-T2D IPFP-ASCs, leading to uncertainty about the overall effect of T2D on OA. Future studies will utilize these characterized IPFP-ASC populations co-cultured with osteoblasts, chondrocytes, and synovial fibroblasts to investigate adipose-mediated diabetic complications associated with OA and the impact of diabetes on chondrocyte activity. This can be done using standard transwell cultures and our previously published OA knee microphysiological system [39,62,63].

## 5. Conclusions

The observations reported here suggest that IPFP-ASCs may contribute to an enhanced immune response through the increased expression of *COX-2* and *IL-6* when stimulated with IL-1β or co-cultured with M1 macrophages. Moreover, Pre-T2D IPFP-ASCs have elevated expression of *COX-2* and enhanced production of PGE_2_ under inflammatory conditions. Our data suggest that while IPFP-ASCs may have therapeutic potential due to their anti-inflammatory and differentiation properties, exposure to a pro-inflammatory environment, e.g., *IL-1β* stimulation and diabetic status, may reduce these therapeutic properties.

## Figures and Tables

**Figure 1 cells-11-02367-f001:**
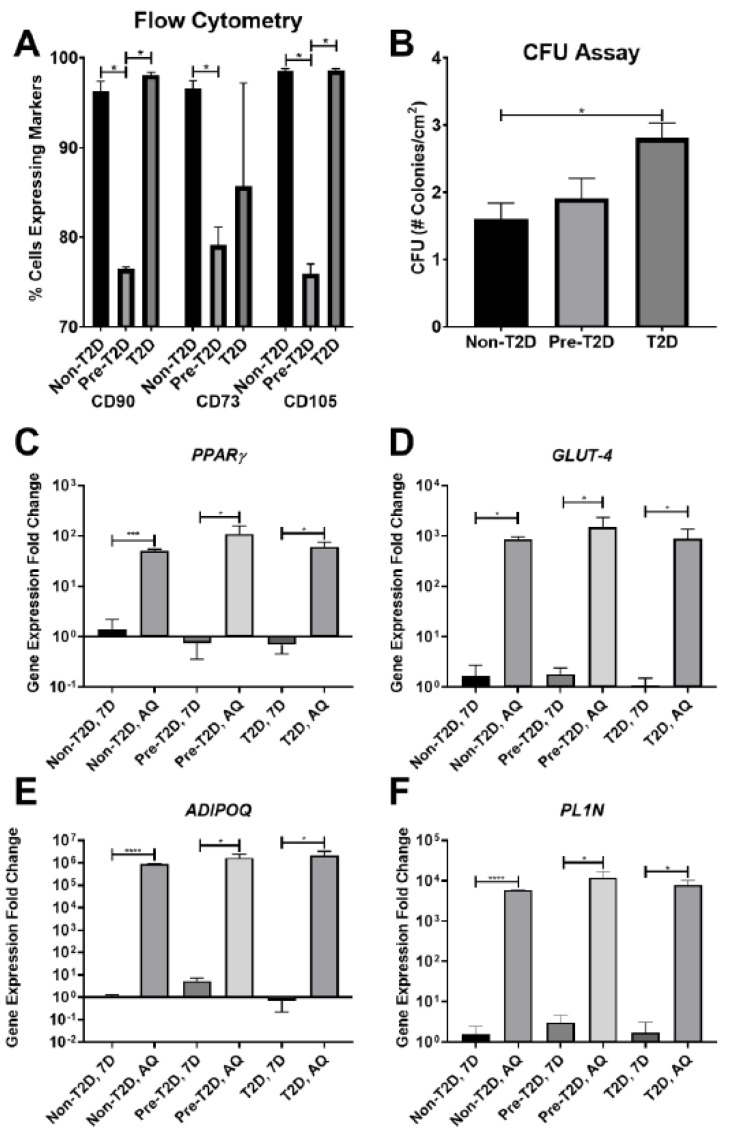
Decreased expression of CD90, CD73, and CD105 on Pre-T2D IPFP-ASCs. (**A**) Flow cytometry for CD90 and CD105 demonstrated decreased expression in Pre-T2D IPFP-ASCs compared to those from Non-T2D and T2D groups (*n* = 3, * *p* < 0.05). (**B**) CFU assay illustrated increased self-renewal properties in T2D IPFP-ASCs compared to Non-T2D (*n* = 3, * *p* < 0.05). (**C**–**F**) RT-qPCR for common adipokines in ASCs and adipocyte differentiated ASCs demonstrated no significant difference in adipogenic potential between Non-T2D, Pre-T2D, and T2D IPFP-ASCs (*n* = 3, * *p* < 0.05, *** *p* < 0.001, **** *p* < 0.0001). Non-T2D IPFP ASCs 7-Day is the control Group set as 1. Non-T2D: IPFP-ASCs from donors without Type II diabetes mellitus, Pre-T2D: IPFP-ASCs from donors with pre-Type II diabetes mellitus, T2D: IPFP-ASCs from donors with type II diabetes mellitus, 7D: Confluent ASCs, AQ: Adipocyte Differentiated ASCs.

**Figure 2 cells-11-02367-f002:**
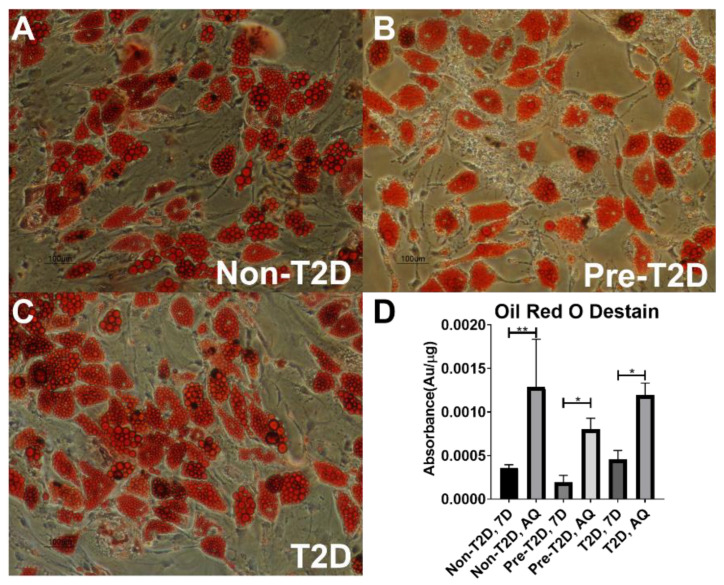
No significant differences in adipogenesis between IPFP-ASCS isolated from Non-T2D, Pre-T2D, and T2D donors. (**A**–**C**) Oil Red O representative image of IPFP-ASC cultured in AdipoQual media for 28 Days (Scale Bar: 100 μm). (**D**) Isopropanol destaining of IPFP-ASCs cultured for 28 days in growth media or AdipoQual Media for 28 days (*n* = 3, * *p* < 0.05, ** *p* < 0.01,). Non-T2D: Non-Type II diabetes mellitus, Pre-T2D: Pre-Type II diabetes mellitus, T2D: Type II diabetes mellitus.

**Figure 3 cells-11-02367-f003:**
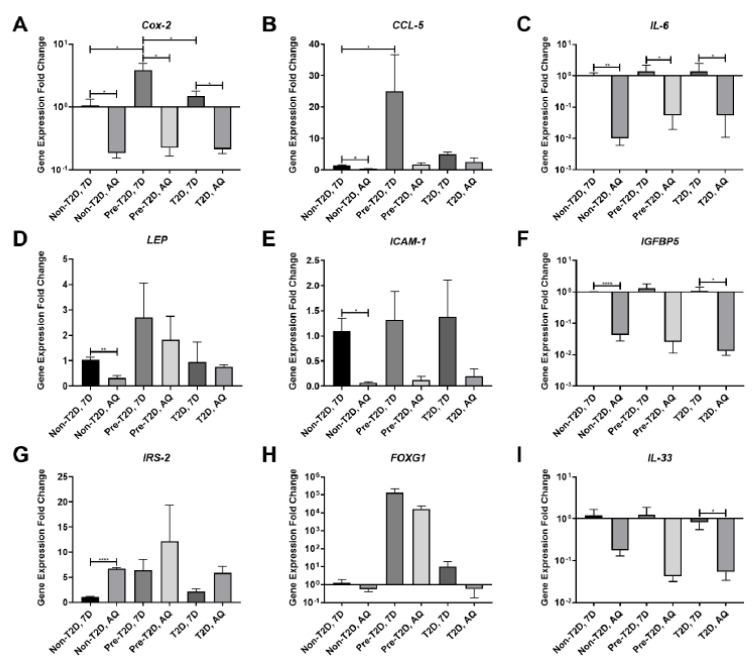
(**A–I**) Decreased expression of pro-inflammatory cytokines by IPFP-ASCs after adipocyte differentiation. (*n* = 3, * *p* < 0.05, ** *p* < 0.01, **** *p* < 0.0001). Non-T2D: Non-Type II diabetes mellitus, Pre-T2D: Pre-Type II diabetes mellitus, T2D: Type II diabetes mellitus.

**Figure 4 cells-11-02367-f004:**
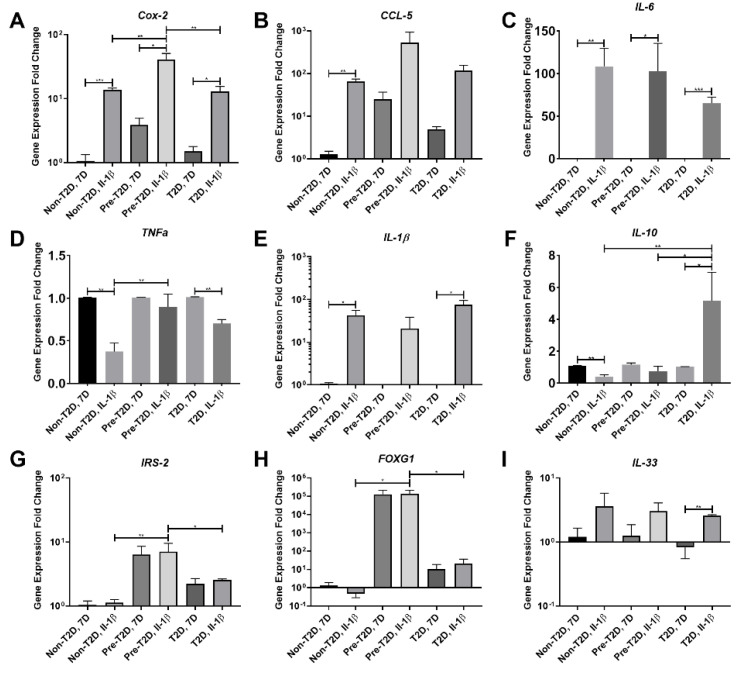
(**A–I**) Increased expression of COX-2 in Pre-T2D IPFP-ASCs compared to Non-T2D and T2D IPFP-ASCs before and after IL-1β treatment. (*n* = 3, * *p* < 0.05, ** *p* < 0.01, *** *p* < 0.001). Non-T2D: Non-Type II diabetes mellitus, Pre-T2D: Pre-Type II diabetes mellitus, T2D: Type II diabetes mellitus.

**Figure 5 cells-11-02367-f005:**
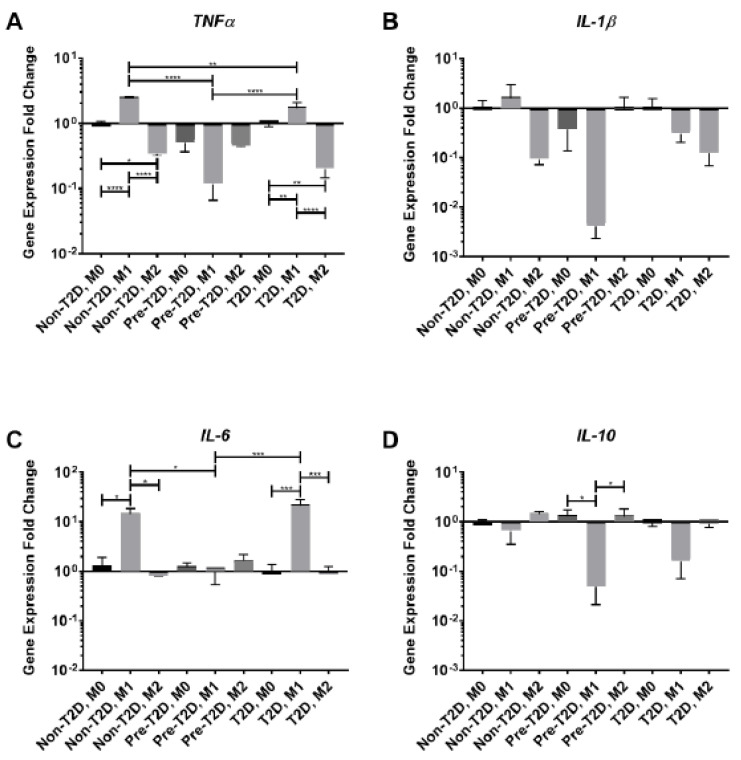
Decreased *TNFα* and *IL-6* expression by M1 macrophages after co-culture with Pre-T2D IPFP-ASCs. (**A**–**D**) Macrophage mRNA expression determined by RT-qPCR (*n* = 3, * *p* < 0.05, ** *p* < 0.01, *** *p* < 0.001, **** *p* < 0.0001). Control Group: M0 Macrophages Co-cultured with Non-T2D-IPFP ASCsNon-T2D: Non-Type II diabetes mellitus, Pre-T2D: Pre-Type II diabetes mellitus, T2D: Type II diabetes mellitus.

**Figure 6 cells-11-02367-f006:**
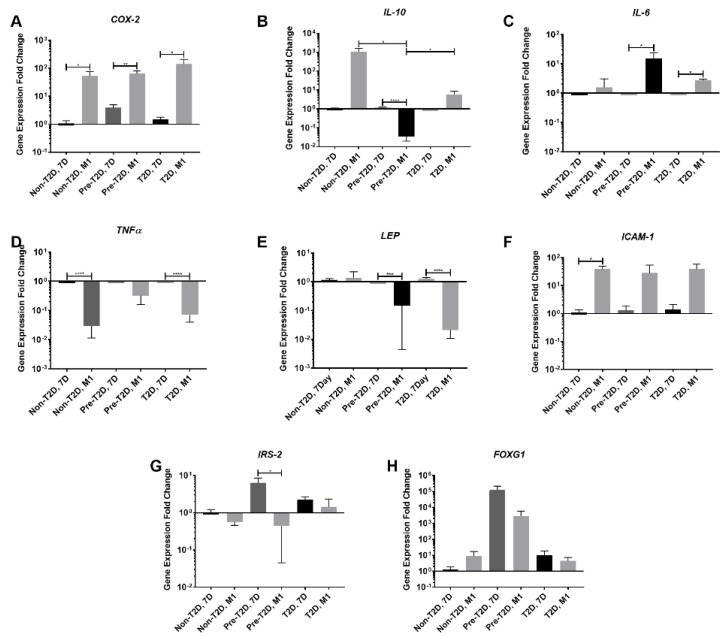
Increased *IL-6* and *Cox-2* expression by Pre-T2D and T2D IPFP-ASCs after co-culture with M1 Macrophages. (**A**–**H**) IPFP-ASC mRNA expression determined by RT-PCR (*n* = 3, * *p* < 0.05, ** *p* < 0.01, *** *p* < 0.001, **** *p* < 0.0001). Reference Group: Non-T2D, 7 Day. Non-T2D: Non-Type II diabetes mellitus, Pre-T2D: Pre-Type II diabetes mellitus, T2D: Type II diabetes mellitus.

**Figure 7 cells-11-02367-f007:**
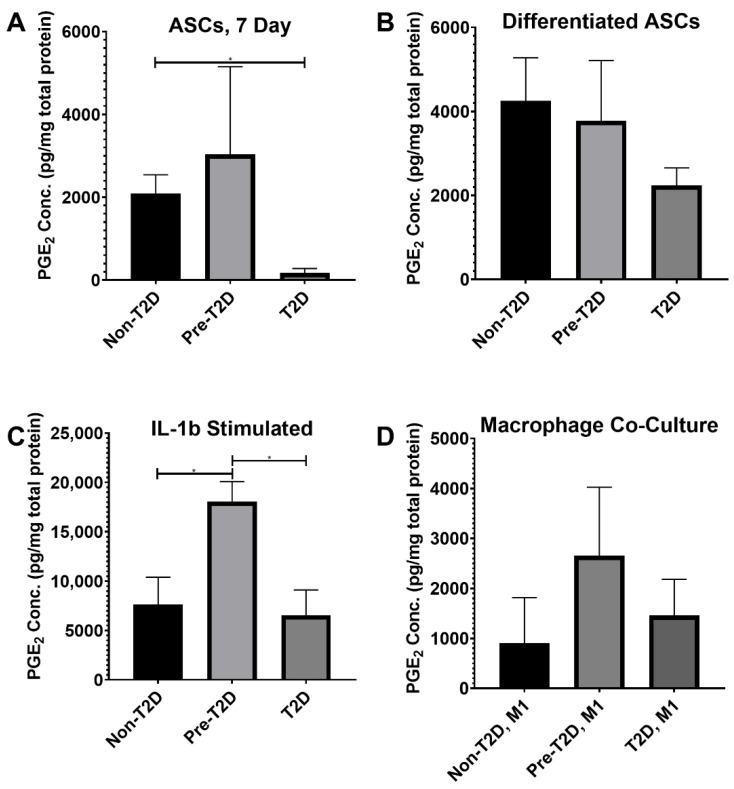
Increased PGE2 concentration in Pre-T2D IPFP-ASC conditioned media after induction with IL-1β. (**A–D**) PGE2 Concentration as determined by ELISA per total protein content as determined by BCA (*n* = 3, * *p* < 0.05). Non-T2D: Non-Type II diabetes mellitus, Pre-T2D: Pre-Type II diabetes mellitus, T2D: Type II diabetes mellitus.

**Table 1 cells-11-02367-t001:** Information on IPFP Donors.

Group	Age	Body Mass Index (BMI)
Non-Diabetic (Non-T2D, *n* = 3)	65.7 ± 8.7	32.1 ± 9.0
Pre-Diabetic (Pre-T2D, *n* = 3)	65.7 ± 4.50	36.6 ± 6.0
Diabetic (T2D, *n* = 3)	53.3 ± 11.3	32.1 ± 3.6

**Table 2 cells-11-02367-t002:** Primer Sequences for RT-PCR.

Name	Forward (5′-3′)	Reverse (5′-3′)
*GAPDH*	TGGTGCTCAGTGTAGCCCAG	GGACCTGACCTGCCGTCTAG
*PPARγ*	AGGCGAGGGCGATCTTG	CCCATCATTAAGGAATTCATGTCATA
*ADIPOQ*	AACATGCCCATTCGCTTTAC	AGAGGCTGACCTTCACATCC
*LEP*	GAAGACCACATCCACACACG	AGCTCAGCCAGACCCATCTA
*FABP4*	AGCACCATAACCTTAGATGGGG	CGTGGAAGTGACGCCTTTCA
*PL1N*	ACAAGTTCAGTGAGGTAG	CCTTGGTTGAGGAGACAG
*LPL*	GAGATTTCTCTGTATGGCACTG	CTGCAAATGAGACACTTTCTC
*FOXG1*	GGCAAGGGCAACTACTGGAT	CTGAGTCAACACGGAGCTGT
*IRS2*	TCTCAGGAAAAGCAGCGAGG	TCACGTCGATGGCGATGTAG
*IL-33*	GCCTTGTGTTTCAAGCTGGG	CCAAAGGCAAAGCACTCCAC
*ICAM-1*	ACCATCTACAGCTTTCCGGC	CAATCCCTCTCGTCCAGTCG
*IGFBP5*	AAGCCTCCCTCACTCTCCAT	TTCCTCCCCACATCGACTCT
*IL-1β*	TCCCCAGCCCTTTTGTTGA	TTAGAACCAAATGTG
*TNFα*	TCTTCTCGAACCCCGAGTGA	CCTCTGATGGCACCACCA
*CCL-5*	CCCCATATTCCTCGGACACC	TCCTTGACCTGTGGACGACT
*COX-2*	TTGCTGGCAGGGTTGCTGGTGGTA	CATCTGCCTGCTCTGGTCAATCGAA

## Data Availability

Not applicable.

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
