# Peer review of "Aberrant Expression of COX-2 and FOXG1 in Infrapatellar Fat Pad-Derived ASCs from Pre-Diabetic Donors"

_cells, 2022, doi:10.3390/cells11152367_

Round 1
Reviewer 1 Report
Thank you to the authors for putting together an interesting manuscript. I feel the content adds to the literature. I do think that the introduction and discussion could be more targeted to better express the findings.
Some examples in the intro -
Line 42-47 includes background on OA that is not immediately pertinent to this publication and could be condensed or taken out.
Line 61-68 again can be compressed or removed
Line 70-72 is where the introduction of the paper becomes a bit more clear, but there is still extraneous information for the reader.
There is no outlined objective for the paper in the introduction and no hypothesis is clearly stated. This leaves the reader a bit unprepared as they move into the Materials and Methods.
Materials and Methods:
Line 125 - How did you detach/harvest the cells? Accumax or Trypsin? The method can effect surface expression of markers.
Line 141 - 142 - Your method for colony counting does not take into account the size of the individual colonies (except that they have to be over 3mm2). Perhaps a total surface area should be considered?
Line 286, Line 340, Line 345 - please remove the findings that are considered "trends" they do not add to the results
Discussion:
The discussion could use a pointed overhaul. Although you have reviewed a lot of literature, there is quite a bit of extraneous information.
Line 409 - rephrase
Line 412-414 - rephrase (unclear)
Line 430 - you are starting a new paragraph with "finally".
Line 448 - 450 - rephrase and consider reworking the paragraph. You indicate that not many "notable differences" have been found but you have spent a great deal of time/effort before this - highlighting some differences in Pre-T2D ASCs.
Line 459 - 466 - compress this if possible. This is a lot of space to be talking about bFGF when your experiment did not utilize it. This may be better placed into a section of "limitations". Other more targeted limitations should be provided.
Line 468 - 473 - please compress this. The previous groups work on the microJoint is not relevant to the current publication. You can suggest future work to include microJoint (as you have), but this can be done in a more efficient manner.
Author Response
We want to thank the reviewers for their insightful comments and suggestions, which we have addressed in the revised manuscript. All concerns have been discussed. All the requested references have been added to the manuscript. The Introduction, the Results and the Conclusion have been strengthened based on the reviews. In the following sections, we have addressed the comments specific to each reviewer.
Reviewer 1:
Thank you to the authors for putting together an interesting manuscript. I feel the content adds to the literature. I do think that the introduction and discussion could be more targeted to better express the findings.
Some examples in the intro -
Line 42-47 includes background on OA that is not immediately pertinent to this publication and could be condensed or taken out.
Line 61-68 again can be compressed or removed
Line 70-72 is where the introduction of the paper becomes a bit more clear, but there is still extraneous information for the reader.
There is no outlined objective for the paper in the introduction and no hypothesis is clearly stated. This leaves the reader a bit unprepared as they move into the Materials and Methods.
Response: The introduction has been modified.
Materials and Methods:
Line 125 - How did you detach/harvest the cells? Accumax or Trypsin? The method can effect surface expression of markers.
Response: The cells were detached using 0.25% trypsin/1 mM EDTA 0.25% trypsin/1 mM EDTA (Cat #: 25200056; ThermoFisher, Waltham, MA, USA). Our extensive experience using trypsin/EDTA shows that it does not affect the expression of stemness markers on ASCs.
Line 141 - 142 - Your method for colony counting does not take into account the size of the individual colonies (except that they have to be over 3mm2). Perhaps a total surface area should be considered?
Response: Our CFU assay aims to provide a quantitative way of investigating stem cell self-renewal. While total colony area could be of interest, plating the ASCs at this low-density results in the formation of colonies generally derived from a single cell. The number of colonies per area gives us an idea of how many cells were able to self-renew and create a colony.
Line 286, Line 340, Line 345 - please remove the findings that are considered "trends" they do not add to the results
Response: The text has been modified as requested.
Discussion:
The discussion could use a pointed overhaul. Although you have reviewed a lot of literature, there is quite a bit of extraneous information.
Line 409 – rephrase
Response: The discussion has been modified.
Line 412-414 - rephrase (unclear)
Response: The text has been rephrased.
Line 430 - you are starting a new paragraph with "finally".
Response: The text has been corrected.
Line 448 - 450 - rephrase and consider reworking the paragraph. You indicate that not many "notable differences" have been found but you have spent a great deal of time/effort before this - highlighting some differences in Pre-T2D ASCs.
Response: The text has been modified to clarify differences in Pre-T2D that were not found in T2D samples.
Line 459 - 466 - compress this if possible. There is a lot of space to discuss bFGF when your experiment did not utilize it. This may be better placed into a section of "limitations". Other more targeted limitations should be provided.
Response: The text has been condensed and the limitations of this study have been added to the discussion.
Line 468 - 473 - please compress this. The previous groups work on the microJoint is not relevant to the current publication. You can suggest future work to include microJoint (as you have), but this can be done in a more efficient manner.
Response: The text has been corrected.
Reviewer 2 Report
The study in interesting and well-written. In addition, it provides important new discoveries about IPFP tissue, which has recently attracted the attention of researchers.
My comments are as follows:
- Lines 48-49, lines 52-54: Reference should be added.
- The introduction on IPFP deserve to be improved. In particular, it should be better explain that there are studies demonstrating that OA IPFP is inflamed and fibrotic compared to controls supporting that this tissue contribute to OA joint inflammation. Fibrosis might modify its biomechanical response in OA. It has been showed that age has remodeling effects on IPFP and this could impact also IPFP-ASCs etc.
- At the end of the introduction, the authors should report the aim/s of the study rather than the results.
- Line 105: the authors reported that they collected IPFP tissues from patients undergoing total knee arthroscopy. “Total knee arthroscopy” is unclear. Did the authors mean total knee artroplasty? What did the patients had? Did the patients had OA? How were pre-Diabetic patients defined? Inclusion/exclusion criteria should be listed. This part needs clarification.
- Line 109: medications should be specified.
- Line 149: “1% Anti/Anti” is unclear.
- Line 150: “uM” should be corrected.
- Lines 172-175: concentrations used are missing.
- Line 183: the authors wrote that they stimulated the cells with IL-1beta. It is unclear the duration of the stimulation. Normally cells are stimulated for 24 h with IL-1beta.
- Line 202: supplier of the kit should be reported.
- Figure 1: here the authors used “7D” and “AQ” as abbreviations, while in figugre 2d “GM and AQ” abbreviations were used. Could the authors use the same abbreviations to avoid confusion in the reader? Did the authors always evaluate gene expression of the markers 28 days after the differentiation?
- The authors did a lot of real time PCR analysis. Indeed, figures 3,4,5 and 6 report only mRNA expression analysis of several markers. As it is well-known that mRNA expression does not always correspond to the protein expression, the authors should evaluate protein expression of the most interesting genes (e.g. cox2).
- Figure 4D: why there is a decrease of TNF-alpha expression after IL-1beta stimulation?
- Figure 5B: statistical analysis is missing.
- Lines 317-319: the authors described LEP and IL-6 expression citing figure 6A and C. However, LEP expression is shown in figure 6E. In lines 325-326 the authors described cox expression citing figure 6E instead of 6A. The authors should check and correct.
- Lines 326-328: ICAM1 expression was found to be elevated in all the three samples after coculturing the cells with M1 macrophages.
- Line 328: “Also, FOXG1 expression increased in Pre-T2D IPFP-ASCs”. This part should be better described.
- Figure 6H: statistical analysis is missing.
- In the methods, the authors reported also that they differentiated IPFP-ASCs cells to osteogenic lineage and performed Alizarin staining. However, this part is not reported.
- Lines 355-356: the authors reported that only Pre-T2D IPFP-ASCs demonstrated an increase in IL-6 gene expression when co-cultured with M1 macrophages. However, also T2D IPFP-ASCs showed an increase of IL-6 expression (figure 6C). The authors should be precise reporting all the information.
- Lines 358-361: references should be provided.
- Line 367: “total knee arthroscopy for T2D patients” is unclear.
- Lines 422-423 and lines 446-448: These parts should be improved. There is a study showing that these IPFP-ASCs cells isolated from OA patients seems to be primed by the OA inflammatory enviroment (DOI: 10.3389/fcell.2019.00323) as IPFP-ASCs isolated from pre-T2D patients by exposure to chronic low-grade inflammation.
- Line 491: “Institutional Review Board Statement: Not applicable”. This part should clarified. The authros reported that they obtained Stanford Hospital Institutional Review Board approval in the methods. Approval code and date should be stated.
- The authors should check all the figure legends and define all the abbreviations used. Labels of the figures should be checked and consistently used (for example, 7D/GM or 7D/7days).
- I did not find the supplementary materials.
Author Response
Reviewer 2:
The study in interesting and well-written. In addition, it provides important new discoveries about IPFP tissue, which has recently attracted the attention of researchers.
My comments are as follows:
- Lines 48-49, lines 52-54: Reference should be added.
Response: References have been inserted.
- The introduction on IPFP deserve to be improved. In particular, it should be better explain that there are studies demonstrating that OA IPFP is inflamed and fibrotic compared to controls supporting that this tissue contribute to OA joint inflammation. Fibrosis might modify its biomechanical response in OA. It has been showed that age has remodeling effects on IPFP and this could impact also IPFP-ASCs etc.
Response: The text has been modified.
- At the end of the introduction, the authors should report the aim/s of the study rather than the results.
Response: The text has been modified to include the goals of the proposed study.
- Line 105: the authors reported that they collected IPFP tissues from patients undergoing total knee arthroscopy. “Total knee arthroscopy” is unclear. Did the authors mean total knee arthroplasty? What did the patients had? Did the patients had OA? How were pre-Diabetic patients defined? Inclusion/exclusion criteria should be listed. This part needs clarification.
Response: The text has been modified. All of the patients had a total knee arthroplasty. Each patients underwent a total knee replacement due to advanced osteoarthritis. The diabetic status of each patient was determined by their primary care physician, which is typically done through a glucose tolerance test or analysis of A1c levels. Unfortunately, due to HIPPA regulations, the specifics of each patient’s diagnosis are unknown to us, nor do we know how each patient's diabetic status was managed. We were simply permitted to know if they were diabetic, pre-diabetic, or normal. Due to the scarcity of tissues, we did not exclude any offered tissues. Due to the correlation of Age and BMI with OA, those patient demographics were reported in Table 1. Patients' OA status was added as a limitation for this study in lines 464 to 468.
- Line 109: medications should be specified.
Response: Patient treatments are unknown, so this statement was removed.
- Line 149: “1% Anti/Anti” is unclear.
Response: It is an antibiotic/antimycotic mixture added to cell culture media to prevent contamination. It has been described in line 121.
- Line 150: “uM” should be corrected.
Response: The text has been corrected.
- Lines 172-175: concentrations used are missing.
Response: Concentrations have been added.
- Line 183: the authors wrote that they stimulated the cells with IL-1beta. It is unclear the duration of the stimulation. Normally cells are stimulated for 24 h with IL-1beta.
Response: The methods have been updated to reflect the 24hr induction period.
- Line 202: supplier of the kit should be reported.
Response: The kit supplier has been added.
- Figure 1: here, the authors used “7D” and “AQ” as abbreviations, while in figure 2d “GM and AQ” abbreviations were used. Could the authors use the same abbreviations to avoid confusion in the reader? Did the authors always evaluate gene expression of the markers 28 days after the differentiation?
Response: The gene expression was evaluated at 7 days (7D) and 28 days.
- The authors did a lot of real time PCR analysis. Indeed, figures 3,4,5 and 6 report only mRNA expression analysis of several markers. As it is well-known that mRNA expression does not always correspond to the protein expression, the authors should evaluate the protein expression of the most interesting genes (e.g. cox2).
Response: We agree with the reviewer that we need to evaluate the protein expressions and we are currently developing methodologies for looking at protein expression. To evaluate COX-2, we conducted an ELISA assay for PGE2, a downstream product of COX-2 activity.
- Figure 4D: why there is a decrease of TNF-alpha expression after IL-1beta stimulation?
Response: We thank the reviewer for this very astute observation. The expected outcome was to observe increased TNFa expression after IL-1 beta stimulation (as illustrated in Fig 5a). However, ASCs have a well-characterized immunomodulatory effect, and the co-culture of ASCs with other cell types, such as chondrocytes has demonstrated a decrease in overall TNFa expression after IL-1b stimulation1. We believe that autocrine signaling in response to IL-1b stimulation induces the observed decrease in TNF-a expression by the ASCs. Future research will investigate the underlying mechanisms that govern this response.
- Figure 5B: statistical analysis is missing.
Response: The data shown in Fig 5B is not statistically significant (p>0.05).
- Lines 317-319: the authors described LEP and IL-6 expression citing figure 6A and C. However, LEP expression is shown in figure 6E. In lines 325-326 the authors described cox expression citing figure 6E instead of 6A. The authors should check and correct.
Response: The text has been corrected.
- Lines 326-328: ICAM1 expression was elevated in all the three samples after co-culturing the cells with M1 macrophages.
Response: The ICAM1 expression was significantly increased in Non-T2D IPFP-ASCs co-culture with M1 macrophages.
- Line 328: “Also, FOXG1 expression increased in Pre-T2D IPFP-ASCs”. This part should be better described.
Response: The text has been modified.
- Figure 6H: statistical analysis is missing.
Response: The data shown in Fig 5B is not statistically significant (p>0.05).
- In the methods, the authors reported also that they differentiated IPFP-ASCs cells to osteogenic lineage and performed Alizarin staining. However, this part is not reported.
Response: Alizarin Red Staining added to the supplement (Fig. S2)
- Lines 355-356: the authors reported that only Pre-T2D IPFP-ASCs demonstrated an increase in IL-6 gene expression when co-cultured with M1 macrophages. However, T2D IPFP-ASCs showed an increase in IL-6 expression (figure 6C). The authors should be precise reporting all the information.
Response: The text has been corrected.
- Lines 358-361: references should be provided.
Response: A reference has been added.
- Line 367: “total knee arthroscopy for T2D patients” is unclear.
Response:
- Lines 422-423 and lines 446-448: These parts should be improved. There is a study showing that these IPFP-ASCs cells isolated from OA patients seems to be primed by the OA inflammatory enviroment (DOI: 10.3389/fcell.2019.00323) as IPFP-ASCs isolated from pre-T2D patients by exposure to chronic low-grade inflammation.
Response: Thank you for pointing out this excellent publication. We have cited it in the text and the text has been updated.
- Line 491: “Institutional Review Board Statement: Not applicable”. This part should clarified. The authors reported that they obtained Stanford Hospital Institutional Review Board approval in the methods. Approval code and date should be stated.
Response: IRB approval codes and dates were included.
- The authors should check all the figure legends and define all the abbreviations used. Labels of the figures should be checked and consistently used (for example, 7D/GM or 7D/7days).
Response: Figures were reviewed and all appropriate abbreviations were made consistent. All abbreviations are now defined in figure legends.
- I did not find the supplementary materials.
Response: Supplementary figures S1-S3 have been included.
References:
- Manferdini C, Maumus M, Gabusi E, et al. Adipose-Derived Mesenchymal Stem Cells Exert Antiinflammatory Effects on Chondrocytes and Synoviocytes From Osteoarthritis Patients Through Prostaglandin E-2. Arthritis Rheum. 2013;65(5):1271-1281. 10.1002/art.37908
Reviewer 3 Report
Dr. O'Donnell reported a link between osteoarthritis (OA) and type II diabetes mellitus (T2D). It is an interesting and important topic. It is a well-designed study and written manuscript.
Minor suggestions: (1) Osteoarthritis should be present in the title; (2) Are the nine IPFP explants collected from OA patients? (3) if the medications used by the diabetic (T2D) patients were listed in Table 1, it will provide more information to understand the data similarity between non-diabetic (Non-T2D) and T2D donors.
Author Response
(1) Osteoarthritis should be present in the title.
Response: We have opted to focus the mansucript more on the ASCs isolated fromthe IPFP of pre-diabetic, diabetic and control patients. Thus the titile reminas the same.
(2) Are the nine IPFP explants collected from OA patients?
Response: Yes, 3 donors from controlp atients, 3 from prediabetic and 3 diabetic.
(3) if the medications used by the diabetic (T2D) patients were listed in Table 1, it will provide more information to understand the data similarity between non-diabetic (Non-T2D) and T2D donors.
Response: Due to HIPAA and IRB regulations, we are not allowed access to this information and ae unable to include it.
Round 2
Reviewer 1 Report
accept in current form
Author Response
I do not see any comments to respond to for Reviewer 1.
Reviewer 2 Report
The manuscript improved after the revision. However, I have still some questions for the authors.
1) In my previous revision, I asked to the authors to give a better introduction of IPFP and OA. However, this part is still scarce. A better introduction on IPFP and OA IPFP pathological changes is necessary, considering also that the IPFPs used in this paper were from end-stage OA patients.
2) Since the authors did not know the patient’s treatments/therapy, it cannot be excluded that treatments could have influenced IPFP cells. This point should be added as a limitation.
3) Figure 2D: “GM” abbreviation is still unclear to me.
4) I noticed that statistical analysis was missing in Figure 6H (question number 18) and the authors replied to me” Response: The data shown in Fig 5B is not statistically significant (p>0.05).”
Author Response
The manuscript improved after the revision. However, I have still some questions for the authors.
1) In my previous revision, I asked to the authors to give a better introduction of IPFP and OA. However, this part is still scarce. A better introduction on IPFP and OA IPFP pathological changes is necessary, considering also that the IPFPs used in this paper were from end-stage OA patients.
Response: A paragraph has been added to the introduction.
2) Since the authors did not know the patient’s treatments/therapy, it cannot be excluded that treatments could have influenced IPFP cells. This point should be added as a limitation.
Response: Unknown treatments as a limitation of the study has been added to the discussion (lines 474-476).
3) Figure 2D: “GM” abbreviation is still unclear to me.
Response: GM abbreviation, referring to cells grown in growth media, has been replaced with 7D in the figure and the text.
4) I noticed that statistical analysis was missing in Figure 6H (question number 18) and the authors replied to me” Response: The data shown in Fig 5B is not statistically significant (p>0.05).”
Response: We apologize for this mistake. The data shown in Fig 6H is not statistically significant (p>0.05).